



# Probabilistic Design of Wind Turbine Blades with Treatment of Manufacturing Defects as Uncertainty Variables in a Framework

Trey W. Riddle[1], Jared W. Nelson[2], and Douglas S. Cairns[3]

[1]Sunstrand, LLC, Louisville, KY
[2]SUNY New Paltz, Division of Engineering Programs, New Paltz, NY
[3]Montana State University, Dept. of Mechanical and Industrial Engineering, Bozeman, MT

*Correspondence to*: Jared W. Nelson (nelsonj@newpaltz.edu)

**Abstract.** Given that wind turbine blades are such large structures, the use of low-cost composite manufacturing processes and materials has been necessary for the industry to be cost competitive. Since these manufacturing methods can lead to

inclusion of unwanted defects, potentially reducing blade life, the Blade Reliability Collaborative tasked the Montana State University Composites Group with assessing the effects of these defects. Utilizing the results of characterization and mechanical testing studies, probabilistic models were developed to assess the reliability of a wind blade with known defects. As such, defects were found to best be assessed as design parameters in a parametric probabilistic analysis allowing for establishment of a consistent framework to validate categorization and analysis. Monte Carlo simulations were found to

adequately describe the probability of failure of composite blades with included defects. By treating defects as random variables, the approaches utilized indicate the level of conservation used in blade design may be reduced when considering fatigue. In turn, safety factors may be reduced as some of the uncertainty surrounding blade failure is reduced when analysed with application specific data. Overall, the results indicate that characterization of defects and reduction of design uncertainty is possible for wind turbine blades.

## 1 Introduction

As part of the Department of Energy sponsored, Sandia National Laboratory led, Blade Reliability Collaborative (BRC), a metric has been developed to precisely address the geometric nature of flaws based on statistical commonality in blades (Nelson et al., 2017). The function of the Flaw Characterization portion of this program has been to provide quantitative analysis for two major directives: acquisition and generation of quantitative flaw data describing common defects in composite

wind turbine blades; and, development of a flaw severity designation system and probabilistic risk management protocol for as-built flawed structures (Cairns). To meet these directives, the effects of porosity, In-Plane (IP) waves, and Out-of-Plane (OP) waves were investigated based on priorities provided by the wind turbine industry (Riddle et al., 2011). In all cases, mechanical testing of flawed laminates was performed and failure strengths/strains were correlated to the characteristic flaw parameter. An example of the correlation between a defect parameter and the composite mechanical response for In-Plane

waves is shown in Figure 1 (Riddle et al., 2012).



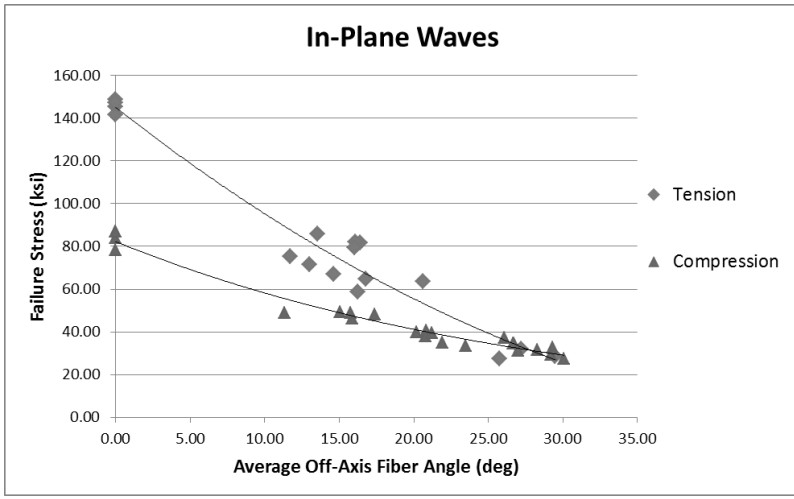

**Figure 1: Individual and trending failure stress for each average off-axis fiber angle tested.**

The typical procedure for certification of wind turbine blades is to use deterministic "Safety Factors" (SF) and apply them to uncertainty variables such as loads, material properties, manufacturing scale up, and manufacturing defects. This is sometimes
blended with statistical treatment for variables such as material allowables, but it does not provide any quantifiable reliability. The SF will generally have some basis from testing, analysis, or experience. The goal of this is to capture "unknown-unknowns" in a conservative manner to minimize failures. However, the amount of conservatism is unknown. Furthermore, if a variable is not correctly considered, the approach may not even be conservative. With a statistical treatment, probabilistic data outside of the database can be accommodated. While the probability of failure for such data may be low, it still exists, as
seen by premature failures associated with manufacturing defects. With a probabilistic approach, overly conservative SF may be decreased, resulting in more reliable blades, at a lower cost (more optimal designs). This would be a new paradigm in the development of certification of wind turbine blades.

This approach has the additional advantage in that the reliability can be quantified as opposed to simply assuming the safety factor will accommodate all unknowns. While it is difficult to make a one-to-one comparison between this standard analysis
technique and the proposed probabilistic approach, important comparisons illustrate the advantage of this method of analysis. To help ensure adequate wind blade design life, which is one intent of the BRC, probabilistic models were developed and analysed.

## 2 Methods & Model Setup

### 2.1 Background Theory

Variations in the structural behavior of composites cannot adequately be characterized by traditional deterministic methods that utilize safety factors to account for uncertain structural response. Moreover, lightweight composites materials are known





to be sensitive to fatigue and defects/damage. Therefore, a methodology focused on reliability targets, which incorporates probabilistic modeling, is essential to accurately determine the structural reliability of a composite structure. Typically, these methods are used with limit state equations in the design process to describe the reliability or probability of failure in a wide variety of a systems (Rackwitz and Flessler, 1978; Ditlevsen and Madsen, 1996; Mahadevan, 2000; Kim et al., 2012) such as

off-shore structures (Kolios and Brennan, 2009). Since a wind turbine blade is a complicated composite structure where uncertainty exists at many levels, each uncertainty variable (e.g. $E$, $G$, $v$, flaw magnitude, and location) can be prescribed a distribution that describes the frequency of occurrence for values of that parameter. These distributions may then be used in the limit state equation to address the total uncertainty or probably of failure in the system.

## 2.2 Model Overview

Previous research has shown the utility of quantifying the influence of defects in composite laminates (Riddle et al., 2013; Dowling, 2012; Samborsky et al., 2012; Nijsssen, 2011; Lin and Styart, 2007). Furthermore, probabilistic design can be accommodated, but has not been adopted as the common approach for wind turbine certification (FAA, 2011). The present work builds upon the work by Nelson et al. (2017) where defect types are classified by known types. The influence of those defects in terms of durability and damage tolerance are determined on a probabilistic basis. This is the basis of the high

reliability of manned aircraft structures. Clearly, the wind turbine industry cannot afford to implement the rigor of FAA FAR 25.571, but elements can be captured to develop quality guidelines, to reduce scrap rate, and to better enable a successful life cycle for composite wind turbine blades. Bacharoudis and Philippidis (2013) presented a similar framework concerning the probabilistic reliability assessment wind turbine rotor blades in ultimate loading. However, the presence of defects was not considered nor was the failure criterion developed based on the fatigue life of composite material subjected to variable loads.

Alternatively, other work has focused on treating the wind loading as variable in the assessment of fatigue life (Hu et al., 2016) and the affect uncertainties in constituent properties have on the stiffness properties of a wind blade (Mustafa et al., 2015).

As noted above, the overall effort can be divided into two major directives: (1) acquisition of relevant defect statistics and defect laden lamina response and (2) development of a probabilistic model to assess the global structural response, probability of failure, and estimation of time to failure for wind blades with flaws. Both directives are addressed within the

context of the framework proposed called the Probabilistic Reliability Protocol (PReP). A conceptual flow diagram showing the interconnectedness of each element of PReP is shown in Figure 2. The PReP algorithm combines defect characterization and probabilistic structural reliability analysis with field and manufacturing data in an iterative feedback loop. A comprehensive reliability program aimed at assessing as-built structures can be divided into four interrelated components and described generally as follows:

a) Effects of Defects: Involves the identification, characterization and analysis of defects. Develops characteristic parameters, material properties and damage models.

    b) Probabilistic Analysis: A stochastic approach that considers multi-scale mechanical property variability, damage/defect detection, residual strength analysis, global, and macro structural response.



    c)    Criticality Assessment (CA): Developed as a surrogate model for the afore mentioned stochastic analysis. It is a time efficient metric for use by operators, manufactures and repair technicians to evaluate the risk of operating a structure with known flaws and/or damage.

    d)    Reliability Estimation and Evaluation: Use of the CA to assess structures on the manufactures floor and in the field. Results from inspections as to the accuracy of the models and the implications to blade reliability are then fed back into the design and evaluation procedures.

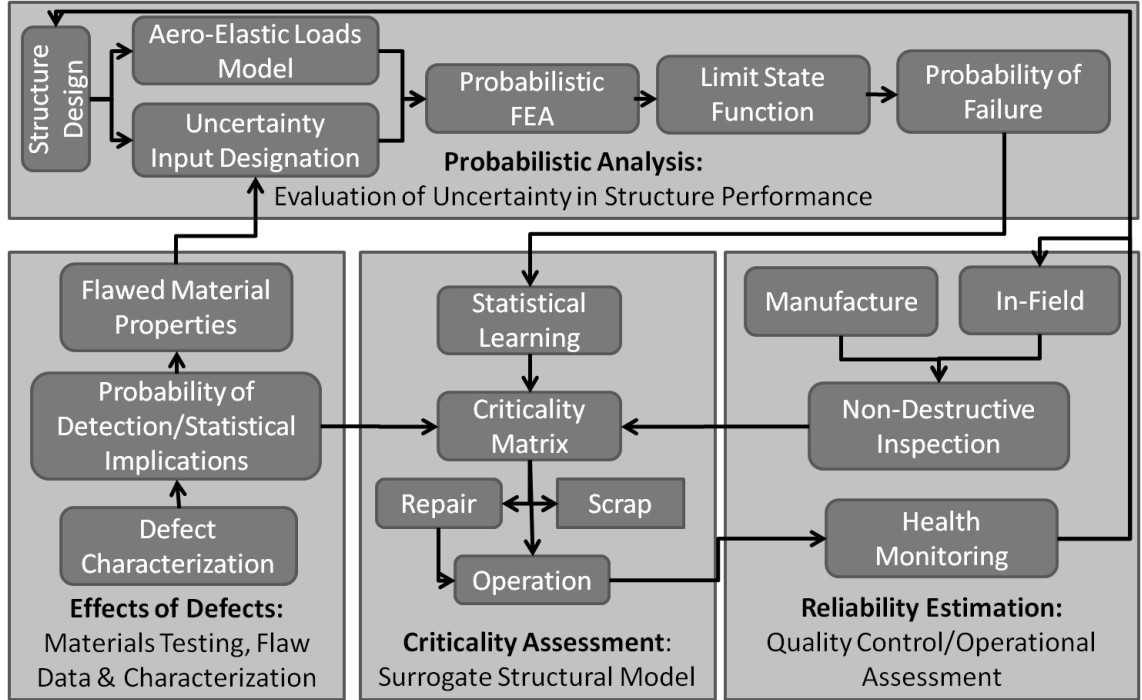

**Figure 2: Conceptual flow diagram of Probability Reliability Protocol (PReP) frame work.**

Each one of the components are complicated and require independent steps which coalesce into the larger framework; however, they may also be utilized independently. Component A: Effects of Defects was the target topic of the preceding paper in this series. This paper will focus on describing the elements of Component B: Probabilistic Analysis as an independent formulation. The general approach, which incorporated finite element simulation into a probabilistic reliability evaluation, adhered to the following steps:

    1)    Build parametrically defined blade model

    2)    Define Random Variables (RV) and their distributions

    3)    Define outputs variables of interest

    4)    Define load scheme

    5)    Perform simulations

    6)    Extract relevant probabilistic output response data



    7)    Input data into reliability analysis

This methodology may be utilized for any application; however, the specifics may vary according to the structure and objectives of the analysis. Table 1 lists the steps necessary to perform the analysis outlined in PReP for a wind blade application. In this table, a title for each step and task are given as well as a short description of the task. Figure 3 illustrates the flow of information and interconnections of the various analysis components. Several of the steps identified in the previous table are notated by the corresponding step number on the figure (Riddle et al, 2013).

**Table 1: Structural reliability analysis hierarchy**



| Step # | Step Name | Description |
|---|---|---|
| **1** | **Analysis Article Set Up** | |
| 1.1 | Article Designation | Establish article of interest |
| 1.2 | Environmental Conditions | Wind speed distribution |
| 1.3 | Governing Article Parameters | Operational Parameters: Tip Speed, RPM, Operating Hours, Design Life |
| **2** | **Structural Analysis** | |
| 2.1 | Finite Element Model | 3-D Shell elements with as-built material properties and layup |
| 2.2 | Flaw Location Discretization | Selection of elements for nodal solution of mechanical response |
| 2.3 | Load Introduction | Uniform pressure distribution applied to HP side of blade |
| **3** | **Development of Failure Criteria** | |
| 3.1 | Fatigue Properties | $\varepsilon$-N Curve for specific R ratios |
| 3.2 | Constant Life Approximation | Piecewise Linear Approximation |
| 3.3 | Designation of Spectrum for Load Reversals | Standardized WISPER reversal spectrum for wind blade loading |
| 3.4 | Derivation of Total Fatigue Cycles | Based on operational parameters |
| **4** | **Flaw Data Implementation** | |
| 4.1 | Development of Flaw Distributions from data | Collected data on waves angles fit to Normal distribution w/non-zero mean. |
| 4.2 | Designation of simulated flaw distributions for comparative analysis | Analyst generated Normal distribution for waves & porosity w/zero mean and for porosity w/non-zero mean |
| 4.3 | Development flaw occurrence distribution | Spatial distribution describing the probably of a flaws existing by location |
| 4.4 | Treatment of flaw structural performance in fatigue | Modification to $\varepsilon$-N Curve single cycle intercept with knockdown factor based on flaw magnitude |
| **5** | **Model verification/tuning** | |
| 5.1 | Model Implementation | Structural model and fatigue failure criteria used on test article |
| 5.2 | Development of Baseline "Design" Case | Load application (pressure) tuned to elicit a blade failure at 20 years (without flaws) |
| **6** | **Probabilistic Analysis** | |
| 6.1 | Probability of Failure | Calculated for all locations for each analysis case. Compared to baseline to how conservatism |
| 6.2 | Time to Failure | Calculated for regions of interest (locations high Pf). |

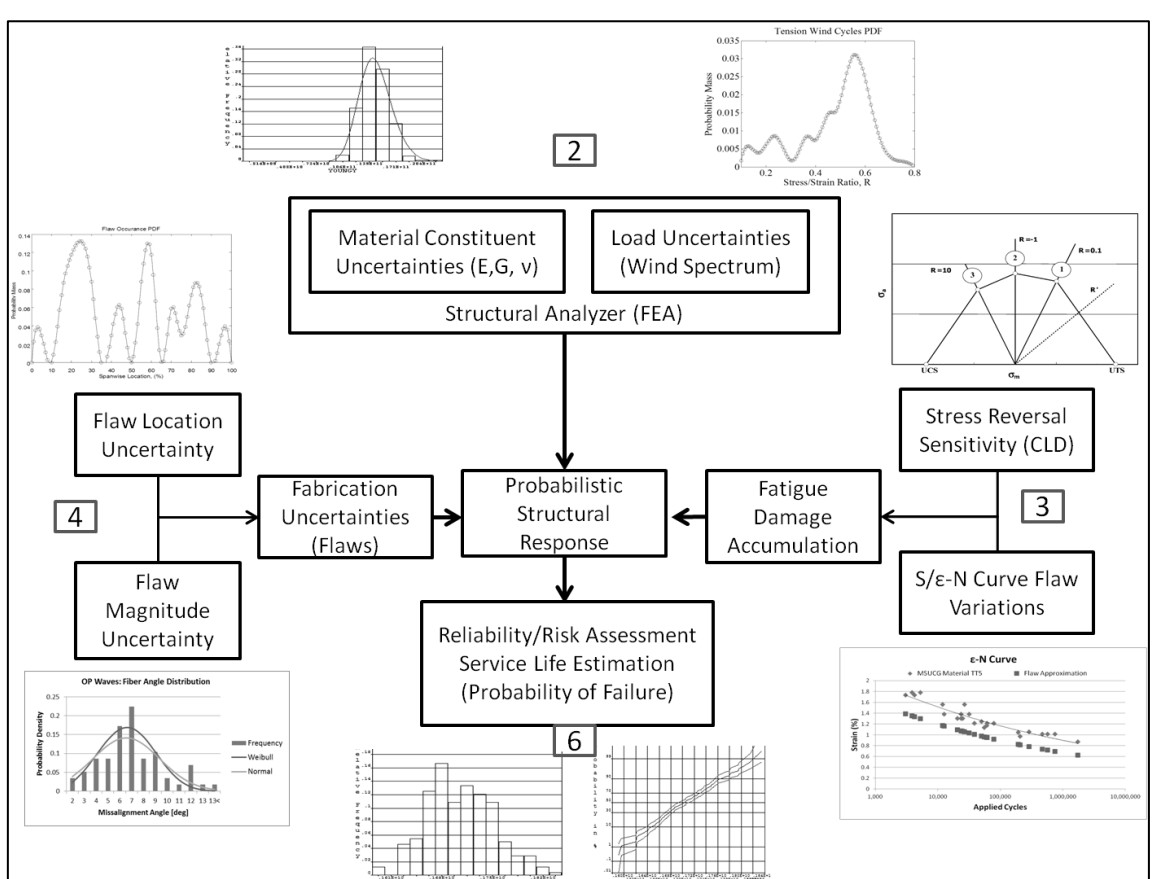

**Figure 3: FEA and Risk Analysis Overview**

## 2.3 Definition of a Performance Function

The overall structural system is a function of a combined Cumulative Distribution Function (CDF), *F*. For this case, a
multivariate probability density function (PDF) is formed as generalized by Equ (1):

$$F(x_1, \ldots, x_n) \equiv Pr(X_1 \le x_1, \ldots X_n \le x_n) \tag{1}$$

The PDF describes how the overall system reacts to the combination of relevant variables. The system reaction to any one
variable can be found by taking the partial derivative of the joint CDF with respect to each of the variables as shown in Equ
(2):

$$f(x) = \left. \frac{\partial^n F}{\partial x_1 \ldots \partial x_n} \right|_x \tag{2}$$

The focus of reliability estimation is typically to describe probability of failure. However, the context of failure varies for each
application. In a damage tolerant design, one might be interested in the probability of failure between the current evaluation
and the next inspection interval. This type of analysis has worked well for the aviation industry where an aircraft can be pulled



into a hanger and inspected relatively easily. A wind turbine blade on the other hand will remain at 100 meters where inspection procedures (and results) are limited. Therefore, the typical design approach is based on a safe life criterion. While an extreme event plays a role in the sudden onset of damage, failure modes are typically considered to be fatigue driven.

Wind is variable, and thus, the resulting bending moment that a blade experiences is variable. Application of an infinitely variable loading scenario to design and test is unreasonable; therefore, rainflow counting is typically used to convert a spectrum of wind speeds (realized structurally as moments) into a set of cycles. The fatigue life can then be used in conjunction with the Palmgren-Miner's rule for linear damage accumulation (Equ (3)) (Dowling, 2012).

$$D = \sum_{i=1}^{k} \frac{n(S_i)}{N(S_i)} \tag{2}$$

where $D$ is the cumulative damage, $n$ is the number of load cycles at the applied stress $S_i$, and $N$ is the number of cycles to failure at $S_i$. Fatigue failure is typically defined to occur when $D$ exceeds a value of 1. A commonly used model for the fatigue life of composites is the power law as described in Equ (4) and modified equation for flaw fatigue life is presented in Equ (5) (Samborsky, 2012; Nijssen, 2011):

$$S = AN^b \tag{4}$$

$$S = KAN^b \tag{3}$$

where $S$ is the maximum applied stress (or strain) N is the number of fatigue cycles, $A$ is the power lower fit coefficient (often referred to as the single cycle intercept), $b$ is the fit parameter for the power law slope, and $K$ is the newly appointed flaw knockdown factor.

Fatigue data of composites containing flaws found in wind turbine blades is not readily available. However, previous studies on damaged composites have shown that the fatigue life slope remains largely unchanged with damage (Lin and Styuart, 2007). Therefore, an idealized approach has been taken to adjust existing material data S-N (or ε-N) curves by a shift in the static failure values (knockdown factor) applied to the single cycle intercept $A$ in Equ (5). Flaw knockdown factors, derived from empirical testing (Nelson et al., 2017), were utilized for this analysis is a scalar quantity used to reduce a material property as a function of the defect characteristic parameter. Presented in Figure 4 is an example of the correlation between knockdown factor and composite mechanical response. An illustration of Equ (5) for a flaw that resulted in a 25% reduced static strain to failure is shown in Figure 5.

The natural extension to this discussion is then to translate a design life of years into cycles. In doing so one can construct the compact limit state function shown in Equ (6):

$$g(X) = 1 - D(X) = 1 - \sum_{i=1}^{k} \frac{\Delta n(\varepsilon_i)}{N(\varepsilon_i)} \tag{6}$$




wherein the resulting strain ($\varepsilon_i$) is a function of the uncertainty parameter vector **X**. This formulation is capable of modeling any fatigue loading spectrum and it has the flexibility to predict failure as a function of applied cycles. Traditional wind turbine design assumes standardized wind loading circannual distribution. Based on this estimation, the performance function can be evaluated two ways: assessing the probability of failure for a specific design life (e.g. 20 years), or assessing the time to failure

5    based on an acceptable probability of failure value. Variations to the analysis to accommodate both predictions are minor and both approaches will be presented.

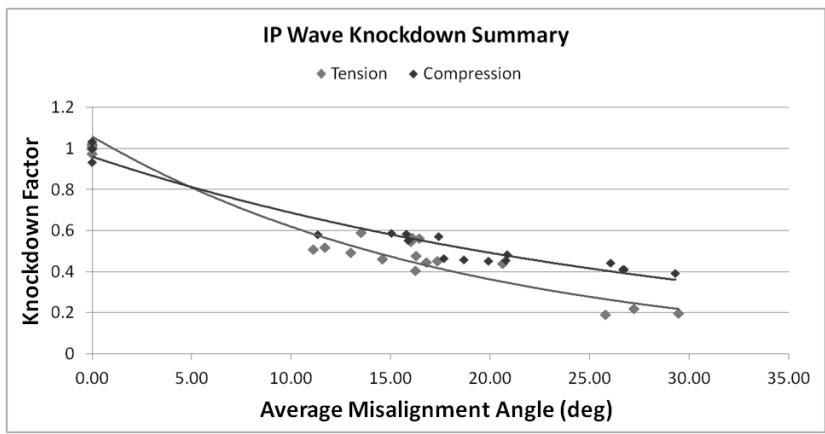

**Figure 4: Empirically derived knockdown factor as related to fiber misalignment angle.**

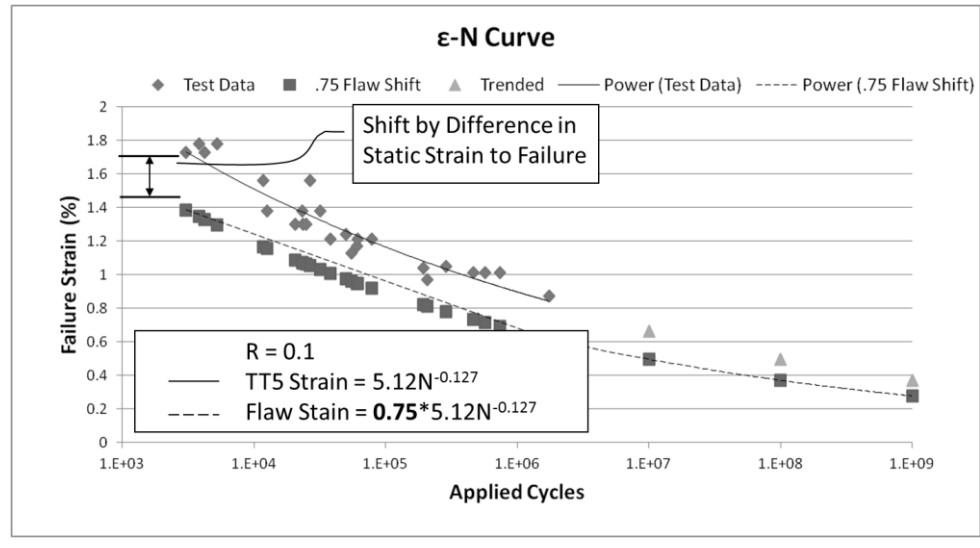

**Figure 5: Representative shifted S-N curve associated with knockdown factor.**





### 2.4 Construction of Simulation

Previous work has shown that composites are sensitive to the variations in loading rehearsal and therefore accurate modeling of fatigue damage accumulation requires usage of fatigue life estimations for specific R-ratios. Constant Life Diagrams (CLD) are used for this purpose, such as the example shown in Figure 6 (Mandell et al., 2010). The amount of data necessary to

5   generate a CLD is often prohibited by cost and time constraints. Therefore, several predictive algorithms have been developed in lieu of copious amounts of testing. Fatigue data for the material systems used in the analysis presented was only available for R=10, R=0.1; Ultimate Tension Strain (UTS) and Ultimate Compression Strain (UCS). The Piecewise Linear methodology (Figure 7) has shown good accuracy in predicting fatigue life with limited amount of test data, therefore it was used (Philippidis and Vassilopoulos, 2004). This method requires a limited amount of test data and performs linear interpolation between the

10   known data points.

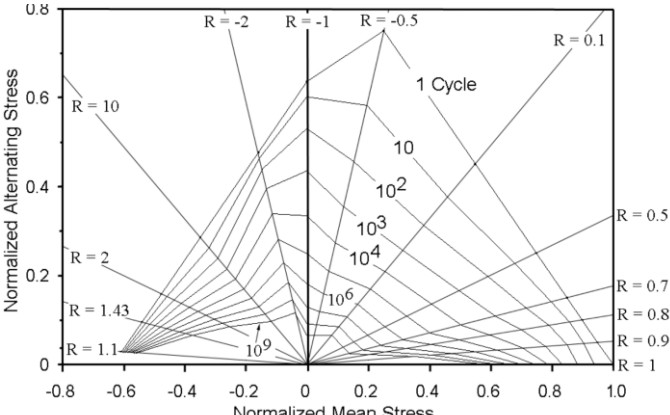

**Figure 6: Representative GFRP Constant Life Diagram.**

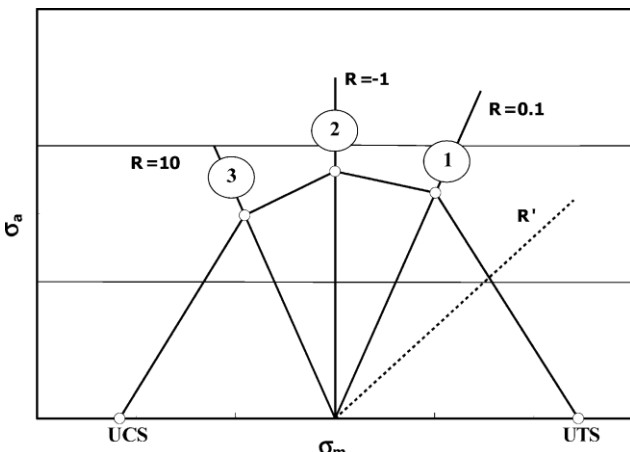

**Figure 7: Approximate Constant Life Diagram represented as a Piecewise Linear Function**



The wind loading spectrum utilized for this analysis was derived from the well-known WISPER load reversal probability distributions (Tenhave, 1992). Two Probability Mass Functions (PMF) were developed from the WISPER data to assess the high and low pressure sides of the blades independently. The high pressure side was assumed to be in tension at all times; thus, the PMF R-values varied from 0.1 to 0.8. Conversely, the low pressure side was assumed to be in compression; thus, R values varied from 1.25 to 10. Based on the WISPER data and these modifications, probability values were generated for 100 discrete load reversal bins. The Probability Mass Distribution and complimentary Cumulative Distribution for the high pressure side are displayed in Figure 8. Typical Computational Fluid Dynamics and Aeroelastic simulations are used to transform these wind speeds into corresponding pressure distributions on the blade surface for use in the structural analysis.

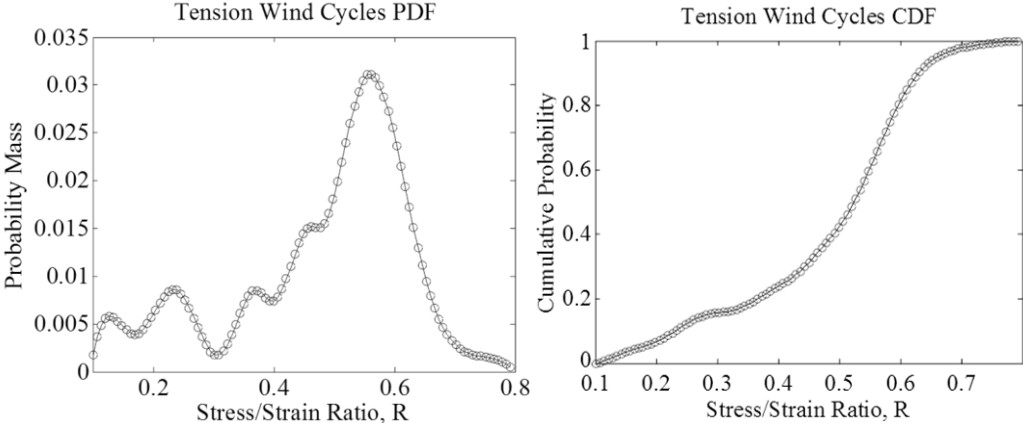

**Figure 8: Wind cycle distributions for high pressure side from WISPER data.**

Wind turbine blades are complex composite structures and one cannot properly assess the integrity of any portion without considering the global response and load share tendencies. It is well known that 2D shell elements used in 3D Finite Element models are required to capture information such as three-dimension distortions, stress concentrations, and buckling strengths. Other methods such as Beam Property Extraction and one-dimensional classical beam section analysis are widely used for preliminary calculations (Veers et al., 1993). These techniques have been used by other investigators for probabilistic analysis of wind turbines (Veers et al., 2003; Lekou and Philippidis, 2005). A full scale blade model (Figure 9) was used in this analysis. The majority of the uncertainty parameter (E, G, ν) variations have been implemented as system wide global properties. The occurrence of flaws has been captured by analyzing and modifying the material properties for a local region of the mesh.

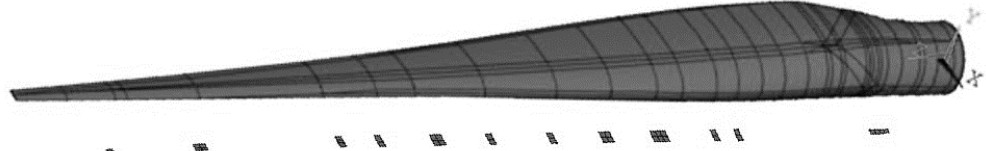

**Figure 9: Finite element model of full blade.**





Flaw locations and magnitude parameters were treated as stochastic variables. First, the probability of a flaw occurring in a specific location was described by a novel spline fit (Figure 10), designating a Probability Mass Function as a function of blade location. One novelty of this approach is the capability for updating procedures that do not rely on the use of traditional,

5    complicated inference techniques. A user, such as a quality control technician, performing inspections on the composite structure may record the frequency and location of observed flaws. These points can then be treated as delta functions in the subsequent piecewise polynomial fitting procedure. Frequencies can easily be updated as more events are recorded enabling the regeneration of distributions used in a statistical analysis. This data is hard to come by therefore a fictitious set of frequencies was selected by the author.  The chosen frequencies and corresponding PMF is displayed Figure 11 and is used in

10   the stochastic analysis to ascertain the probability of a flaw occurring in a specific location. When the sampling algorithm identifies the existence of a flaw, a second distribution describes the probability of the flaw's characteristic parameter magnitude. Figure 12 displays the treatment of an example flaw magnitude as an uncertainty parameter used in this analysis. As noted below, it was found that off-axis fiber angles of waves collected in a survey of wind turbine blades follow typical distributions such as Normal and Weibull.

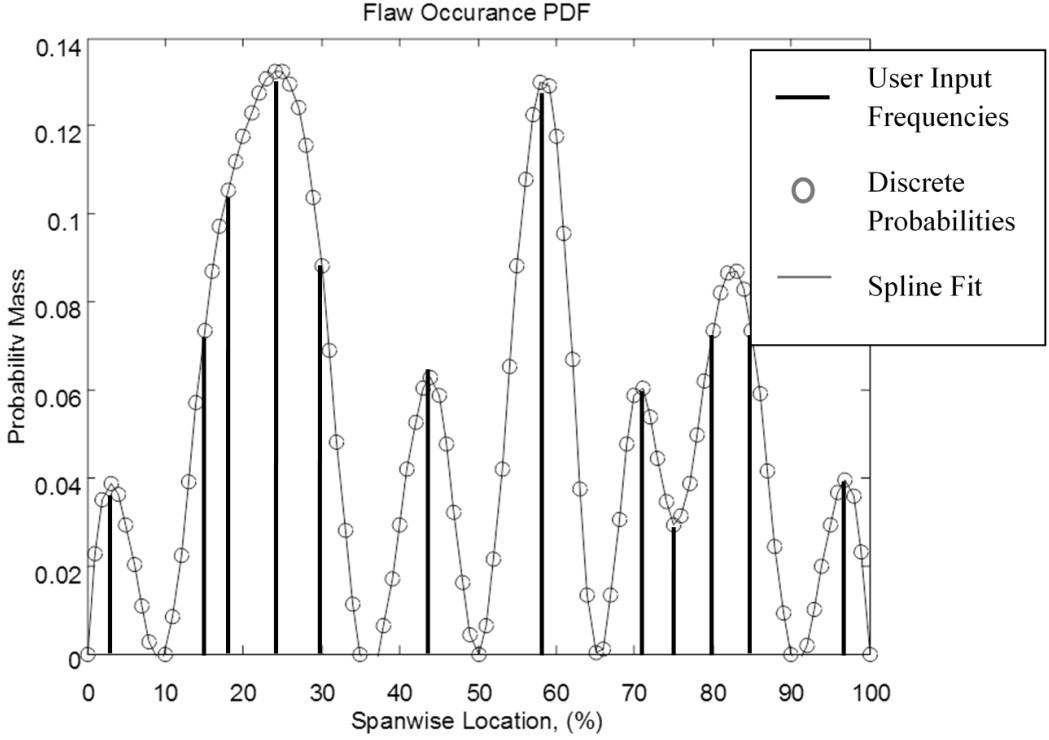

**Figure 10: Probability of flaw mass, or occurrence, spatially distributed along span-wise length.**





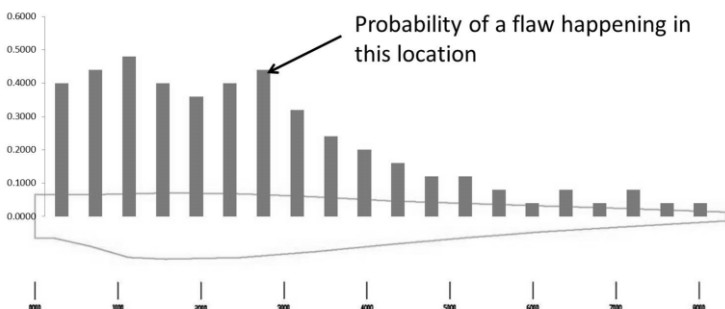

**Figure 11: Probability mass of flaws at each blade location.**

## 3 Case Study

A 9-m wind blade designed by Sandia National Laboratory was used as the article of investigation in this analysis. A 3-D finite
element analysis (FEA) model using shell elements was generated to match the actual blade laminate and planform schedule.
A benchmark standard International Electrotechnical Commission (IEC) approach to fatigue evaluation was used to develop
the baseline analysis, Case 0, to which two probabilistic analyses were then compared. For all analyses the blade spar was
discretized into 100 locations. The maximum nodal strain response in the spar laminate 1 direction (spanwise - material
tension/compression) was output from the FEA model for use in the post processing script. A combined fatigue and
probabilistic analysis was then performed on each location using Monte Carlo simulation. The methodology for each case is
described below with discussion of the results following.

### 3.1 Case 0: Baseline (Design)

Information on the design of the blade article was not readily available; therefore, the blade was reversed engineered to
developed a load scenario which would cause a fatigue failure in 20 years. The designation of an applied pressure load was
considered arbitrary in that it need only provide a referencing point to objectively evaluate analysis techniques. For this
Baseline Case, the IEC Safety Factor Fatigue formulation was used. IEC recommends the usage of traditional linear damage
accumulation employing the Palmgren-Miners rule. The IEC fatigue analysis process can be paraphrased as follows: *Fatigue
damage shall consider effects of both cyclic range and mean strain, and all partial safety factors (load, material and
consequences of failure) shall be applied to the cyclic strain (or stress) range for assessing the increment of damage associated
with each fatigue cycle* (IEC, 2005). Given the relevance to the entire study, the IEC's material Safety Factor ($\gamma_m = 1.3$) was
used with the available material properties test data; therefore, it was the target of the probabilistic analysis.

### 3.2 Case 1: Probability of Occurrence

This analysis case utilized two probability distributions to describe defect uncertainty. The first distribution used in the analysis
is Probability of Occurrence. This distribution describes the probability mass of a flaw existing in a blade using a spatial



distribution. For this analysis, a 1-D cubic spline distribution was used to allow for flaws down the length of the spar cap. The spline formulation allows for high fidelity, continuous interpolation of probabilities between specific locations of known flaw frequencies. When the simulation predicted a flaw's existence, then a second distribution was used to describe the magnitude of the flaw based on the actual field data collected from utility scale wind blades. An example of the distribution and sample

5  set used in the analysis for In-Plane waves is displayed in Figure 12.

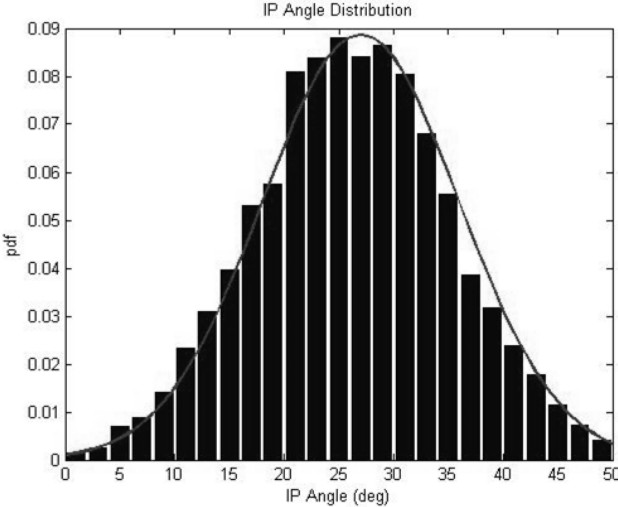

**Figure 12: Distribution of sampled magnitudes of in-plane wave misalignment angles used in Case 1.**

### 3.3 Case 2: Half Gaussian Fiber Wave Magnitude

This analysis case utilizes only one probability distribution to describe defect uncertainty. The analysis assumes that there is a

10  100% chance of a flaw occurring at every location in the blade (Figure 13). The flaw occurrence magnitude is described by a one sided probability distribution (Figure 14). For this case, a flaw magnitude of zero would indicate that there is no flaw at that specific location.

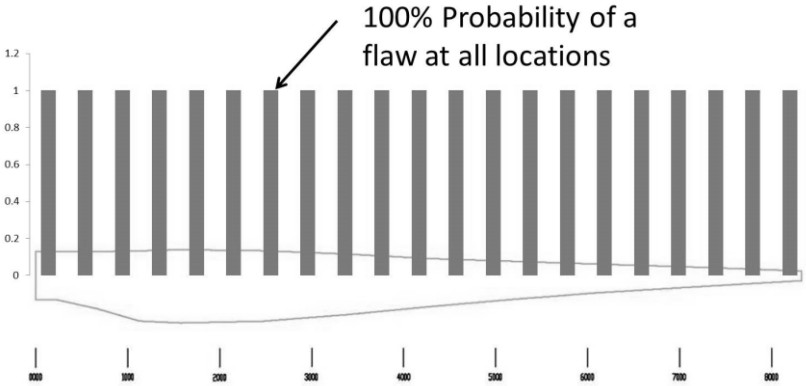

**Figure 13: Probability mass of 100% that a flaw is located at each blade location used in Case 2.**

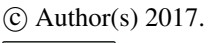


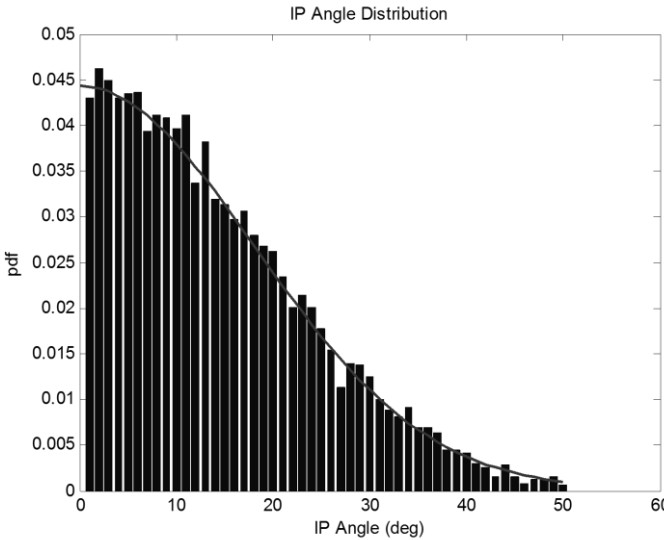

**Figure 14: Half-Gaussian distribution of sample set for flaw magnitudes used in Case 2.**

## 4 Results & Implications

The assumption of a low probability of failure during a typical 20-year blade lifetime was used for this analysis since the probability of failure used for the IEC safety factors was not known. In order to validate the model, this assumption was used for the baseline Case 0 scenario whereas Case 1 and 2 assumes failure within the blade lifetime due to manufacturing flaws not inherent in the certification process. For each case, the likelihood of reaching failure for a given safety factor are presented as failure probabilities which allows for easy comparison. In short, the absolute probability of failure could be tracked through the 20-year lifecycle if the design failure probability is known. As such, both the prescribed IEC and reduced material safety factors were used in the evaluation of both Case 1 and 2 allowing for direct comparison of the level of conservatism. Using Monte Carlo simulations and experimental strain responses, analysis samples and failure probabilities were generated.

### 3.1 Case 1: Spatially Varying Distribution of Defects

For each location of the blade, the FEA simulation calculated strain. Considering the defects as random variables, the Probability of Failure ($P_f$) was then determined for each blade location along the length (Figure 15, left). It may be seen that the blade has a 100% probability of failure at the location 22% along the length of the span. These results were then used to determine the critical point along the blade length by relating $P_f$ to time in service using a linear fatigue damage accumulation model at each location. As seen in Figure 15 right, based on the number of the cycles at the 22% location failure will occur approximately 7 years into the lifecycle. It is important to note that is "worst on worst" with the inclusion of a typical material Safety Factor.



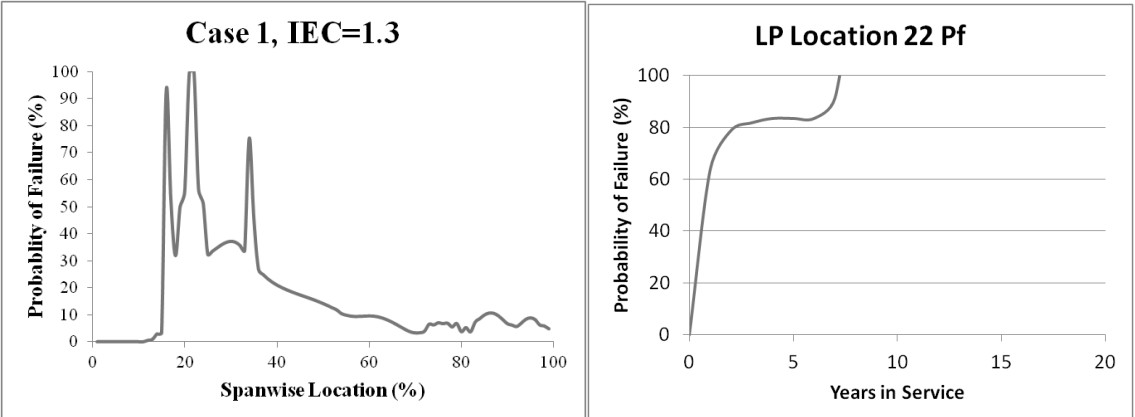

**Figure 15: Probability of failure by location (left) and as a function of time for location 22 (right).**

While this case indicates a significant chance of failure, the blade will likely be overdesigned if a safety factor is used with a probabilistic simulation of defects to ensure a reasonable $P_f$. To quantify this implication, the same model was run with the safety factor reduced to 1.15 from 1.3. As seen in Figure 16, while the locations of the critical points remain the same, none of these points have a 100% $P_f$. As such, these results imply that additional structural reinforcements are not necessary, meaning weight and cost can be reduced. This approach has the added benefit of introducing some level of quantifiable reliability, as opposed to the "assumed to be small probability of failure" of the Safety Factor approach.

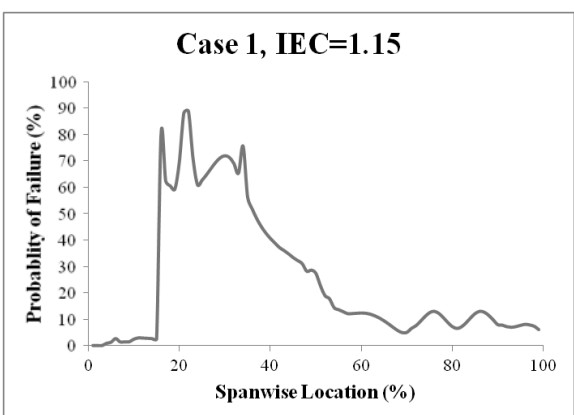

**Figure 16: Probability of failure by location with reduced IEC safety factor.**

**3.2 Case 2: Half Gaussian Fiber Wave Magnitude Results**

As noted, the inputs were then modified using a Half Gaussian distribution (Figure 14) with a 100% probability of a flaw at every location. The case was run for both safety factors and the results are shown in Figure 17. While it is evident that in both cases the $P_f$ approaches 100% failure probability, the reduction of the safety factor results in a reduced estimation that failure will occur which is consistent with the results of Case 1.

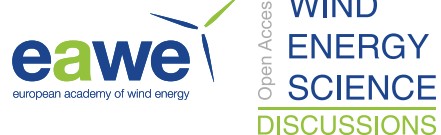

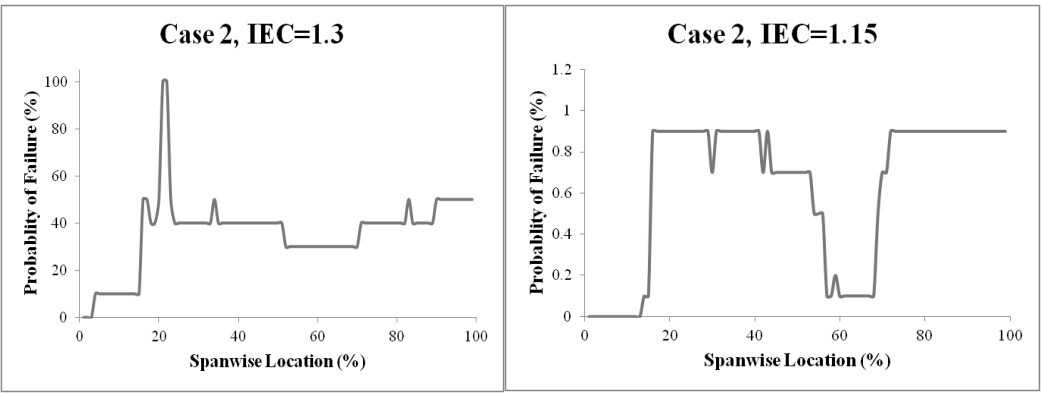

**Figure 17: Probability of failure by location for standard IEC safety factor (left) and reduced IEC safety factor (right).**

### 3.3 Implications of Probabilistic Approach to Reliability

As with any analytical method, detailed and accurate inputs are necessary to use this probabilistic analysis to address
uncertainty of blades with manufacturing defects. When the two cases are compared, it is evident that distributions of flaw
magnitude effect the result significantly as seen when Figures 15 and 16 are compared to Figure 17. The differences are
amplified further when the strength reduction is considered where a dramatic shift in laminate strength is noted as seen in all
four portions of Figure 18. The variations between the two cases are significant and it is clear the impacts on the laminate
when flaws are assumed to be occur at all locations as in Case 2 (Figure 18 c/d). While likelihood of instances with strength
reduction decreases in this case, the reduction of strength is likely to be greater which aligns with previous testing of wavy
laminates that indicated an exponential decrease in laminate strength (Riddle et al., 2012). While this trend is meaningful, it is
imperative to recall that these distributions were generated during this investigation and may not be indicative of the industry
at large or of any one particular manufacturer's process/products. Therefore, it is also imperative that test data representative
of the materials used in the design system be established.



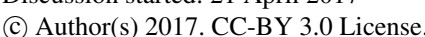



**Figure 18: Strength reduction sample sets as a function of flaw magnitude distributions for: Case 1 in compression (a); Case 1 in tension (b); Case 2 in compression (c); and, Case 2 in tension (d).**

### 3.4 Model Validation via Experiments

To validate these methods, the BRC supported testing of a subscale (9-m) version of a multi-megawatt wind turbine blade that was manufactured with intentional defects. The National Wind Technology Center facilities were used to actuate the blade at three locations allowing for multiple flaws to be assessed individually and with geometric considerations (Figure 19, Figure 20 left). In parallel, the known uncertainty values were used to run three Monte Carlo simulations allowing for direct comparison of the results. As seen in Figure 20 (right), the failure occurred at an out-of-plane flaw as was also predicted by the numerical simulation. Not only did this validate the methods described herein, but this blade scale testing provided insight into the scaling factors and indicated that a local failure constituted a global structural failure.





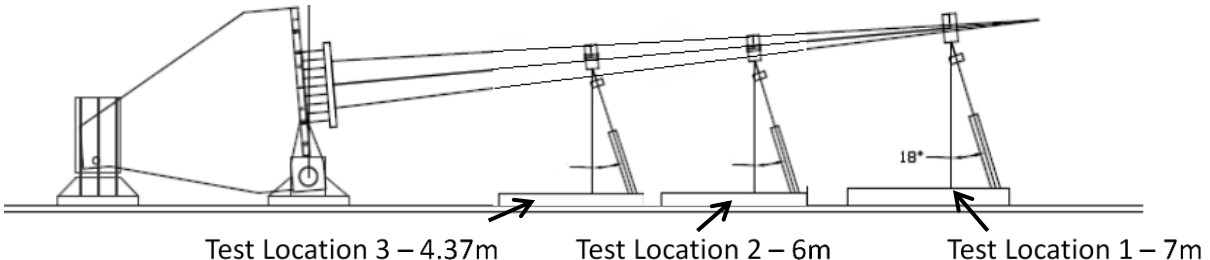

Test Location 3 – 4.37m          Test Location 2 – 6m          Test Location 1 – 7m

**Figure 19: Representation of subscale blade test layout and testing locations.**

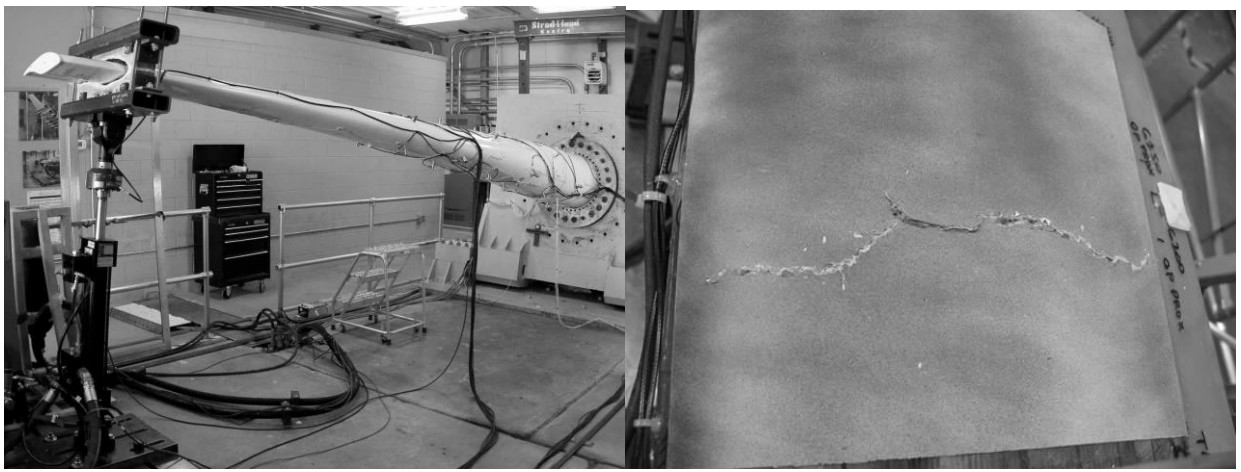

**Figure 20: Actual sub-scale blade test (left) with final failure at flaw location (right).**

## 5   4 Conclusions & Impact on Wind Turbine Blade Designs and Certification

This work postulates that one should quantify and assess manufacturing defects as to their magnitude and criticality for durability and damage tolerance. The same can be said of other important probabilistic distributions affecting reliability where Safety Factors are used in lieu of probabilistic calculations.

The two approaches detailed in this analysis, essentially known defect distributions and blades assumed to have defects, but

10   without any spatial statistic information (e.g. existing fleet) have been demonstrated to show the utility of probabilistic analysis with respect to reducing conservative Safety Factors. Understanding these are critical in terms of reliability is important if one wants to justify reducing Safety Factors. Both magnitude and distribution are important for a comprehensive probabilistic reliability analysis.

The probability analysis needs to be incorporated into a comprehensive program, not just the assessment of a specific defect

15   or probabilistic parameter. A holistic approach to reliability results in a stochastic reliability infrastructure. This aids in the design process as well, with the ability for continuous improvement throughout the product lifecycle. As improvements are



made, Safety Factors can be reduced with the associated impacts on cost. While a full Damage Tolerant Design process is not practical from a cost basis, the approach presented herein has important elements. This includes an inspection program, damage growth laws, and residual strength with defects.

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
