# Peer review of "Effects of defects in composite wind turbine blades - Part 3: A framework for treating defects as uncertainty variables for blade analysis"

_Wind Energy Science, 2017_

## Referee Comment (RC1) · Anonymous Referee #1 · 22 Aug 2017

This paper is presented very well and introduces some really interesting aspects in probabilistic performance characterisation of wind turbine blades. The following minor comments might help to improve the paper: 1. The title of the paper may need a rethink. I have expected further discussions on links between safety factors and design expectations. 2. Some enhanced discussion on target reliability levels and links to blade life time would be helpful. 3. Section 3.4 seems very interesting but limited information is given, is it possible to expand the discussion around Figures 19 and 20? 4. The paper seems to end without many concrete conclusions, is it possible to improve this section? 5. There are some typos in the paper, please recheck the draft manuscript.

---

## Referee Comment (RC2) · Anonymous Referee #2 · 27 Oct 2017

This paper presents the probabilistic design of wind turbine blades considering the manufacturing defects. Overall, the paper is written well, and the work presented in the paper is interesting.

The specific comments are as follows.

1. The quality of figures in the paper needs to be improved through using larger font size and increasing the resolution.

2. The introductory section needs to be expanded. For instance, a review of relevant studies on the probabilistic design of wind turbine blades should be added.
3. More details of the wind turbine blade used in the case study should be given.

4. It would be appropriate to use a table to list all the stochastic variables considered in the study. Additionally, the distribution type, characteristic values, standard deviation of each stochastic variable should be given.

5. It would be appropriate to add a case study to validate the FEA model of the wind turbine blades used in this paper.

6. The target probability of failure for wind turbine blade given by design standards is generally very low. Can authors justify why the calculated probability of failure (e.g. the results presented in Fig. 15) is much higher than the target probability of failure given by design standards?

---

## Author Comment (AC1) · 24 Nov 2017

**Author Comments for Anonymous Referee #1**

The authors are grateful for the comments, suggestions, and insight from the reviewer.  Please find responses below.

**Question #1:** The title of the paper may need a rethink. I have expected further discussions on links between safety factors and design expectations.

> *AR* We are working on a new title. Perhaps "A Framework for Treating Defects as Uncertainty Variables in Wind Turbine Blade Analysis".

**Question #2:** Some enhanced discussion on target reliability levels and links to blade life time would be helpful.

> *AR* The purpose of the article is not to address what a target reliability level should be but rather to propose a methodology wherein the effects of defects can be characterized probabilistically. Some discussion will be provided which characterizes the current approach to blade reliability and how it can be enhanced by this process.

**Question #3:** Section 3.4 seems very interesting but limited information is given, is it possible to expand the discussion around Figures 19 and 20?

> *AR* Originally, the authors planned to submit an entire paper on this section as part of a series. Some additional dialog will be provided to better tie in the two figures and the overall culmination of the body or work.

**Question #4:** The paper seems to end without many concrete conclusions, is it possible to improve this section?

> *AR* The authors are working to better state the conclusions. The work has shown that if wind turbine blades can be designed using a probabilistic approach that incorporates defects, the generic safety factor can be reduced. This will ultimately lead to reduced costs in the construction of blades as they will not need to be "over-built".

**Question #5:** There are some typos in the paper, please recheck the draft manuscript.

> *AR* The authors are proof reading in detail and will address all typos.

---

## Author Comment (AC2) · 24 Nov 2017

**Author Comments for Anonymous Referee #2**

The authors are grateful for the comments, suggestions, and insight from the reviewer. Please find responses below.

**Question #1:** The quality of figures in the paper needs to be improved through using larger font size and increasing the resolution.

> *AR* We are working to recreate many of figures with larger font and increased resolution. This a large task coordinating between the three authors and work that in some cases is several years old.

**Question #2:** The introductory section needs to be expanded. For instance, a review of relevant studies on the probabilistic design of wind turbine blades should be added.

> *AR* Prior studies on the probabilistic design of wind turbine blades have been mentioned in section 2.2. The authors are working to redraft the introduction to better address this prior art.

**Question #3:** More details of the wind turbine blade used in the case study should be given

> *AR* Additional details will be added to describe the 8.325m wind turbine blade with fiberglass spar based on the Sandia Blade System Design Study (BSDS) used in this study. The blade was designed as mechanism to study large scale commercial blade construction at a smaller and more manageable subscale size.
>
> The following reference will also be added:
>
> Berry, D. "Blade System Design Studies Phase II: Final Project Report" *No. SAND2008-4648. Sandia National Laboratories, 2008*.

**Question #4:** It would be appropriate to use a table to list all the stochastic variables considered in the study. Additionally, the distribution type, characteristic values, standard deviation of each stochastic variable should be given.

> *AR* A table will be added.

**Question #5:** It would be appropriate to add a case study to validate the FEA model of the wind turbine blades used in this paper

> *AR* Section 4 describes briefly the physical testing of a subscale wind blade with introduced flaws. More details will be added to reference prior usage of the NUMAD preprocessor and its validation as well as the results of the testing performed in this work, wherein actual strain data collect on the test specimen was consist with FEA model predicted values.
>
> The following reference will also be added:
>
> Resor, B., Paquette, J. "A NuMAD Model of the Sandia TX-100 Blade" *No. SAND2012-9274. Sandia National Laboratories, 2012*.

**Question #6:** The target probability of failure for wind turbine blade given by design standards is generally, very low. Can authors justify why the calculated probability of failure (e.g. the results presented in Fig. 15) is much higher than the target probability of failure given by design standards?

> *AR* Additional details will be provided which describe a background to the failure of probability analysis. The Probably of failure is artificially high as the load case in this analysis was chosen intentionally to yield a fatigue failure of the blade (using a safety factor of 1.3) in 20 years. Using

this as the starting point, a stochastic analysis is performed in addition to using the safety factor and the results compared.

---

## Author Response (AR1)

**Author Comments for Anonymous Referee #1**
The authors are grateful for the comments, suggestions, and insight from the reviewer. Please find responses below.

5    **Question #1:** The title of the paper may need a rethink. I have expected further discussions on links between safety factors and design expectations.

    *AR*   The title has been changed to "Effects of defects in composite wind turbine blades - Part 3: A framework for treating defects as uncertainty variables for blade analysis".

**Question #2:** Some enhanced discussion on target reliability levels and links to blade life time would be helpful.

10     *AR*   The purpose of the article is not to address what a target reliability level should be but rather to propose a methodology wherein the effects of defects can be characterized probabilistically. Some discussion has been provided which characterizes the current approach to blade reliability and how it can be enhanced by this process.

**Question #3:** Section 3.4 seems very interesting but limited information is given, is it possible to expand the discussion around Figures 19 and 20?

15     *AR*   Originally, the authors planned to submit an entire paper on this section as part of a series. Some additional dialog has been provided to better tie in the two figures and the overall culmination of the body or work. An additional refence has been provided where more details can be found.

**Question #4:** The paper seems to end without many concrete conclusions, is it possible to improve this section?

    *AR*   The authors have supplied additional discussion to better state the conclusions. The work has shown that if wind
20     turbine blades can be designed using a probabilistic approach that incorporates defects, the generic safety factor can be reduced. This will ultimately lead to reduced costs in the construction of blades as they will not need to be "over-built".

**Question #5:** There are some typos in the paper, please recheck the draft
manuscript.

25     *AR*   The authors performed additional proof reading and will addressed many typos.

**Author Comments for Anonymous Referee #2**
The authors are grateful for the comments, suggestions, and insight from the reviewer. Please find responses below.

30    **Question #1:** The quality of figures in the paper needs to be improved through using larger font
size and increasing the resolution.

    *AR*   Many figures have either been replaced or modified.

**Question #2:** The introductory section needs to be expanded. For instance, a review of relevant
studies on the probabilistic design of wind turbine blades should be added.

35     *AR*   Prior studies on the probabilistic design of wind turbine blades have been mentioned in section 2.2. The authors
have added some limited dialog on this topic.

**Question #3:** More details of the wind turbine blade used in the case study should be given

    *AR*   Additional details have been added to describe the wind turbine blade.
The following reference was also added:

Berry, D. "Blade System Design Studies Phase II: Final Project Report" *No. SAND2008-4648. Sandia National Laboratories, 2008.*

**Question #4:** It would be appropriate to use a table to list all the stochastic variables considered in the study. Additionally, the distribution type, characteristic values, standard deviation

5  of each stochastic variable should be given.

   *AR*  A table has been added.

**Question #5:** It would be appropriate to add a case study to validate the FEA model of the wind turbine blades used in this paper

10  *AR*  Section 4 describes briefly the physical testing of a subscale wind blade with introduced flaws. More details have been added referencing usage and verification of the code.
   The following reference was also added:
   Resor, B., Paquette, J. "A NuMAD Model of the Sandia TX-100 Blade" *No. SAND2012-9274. Sandia National Laboratories, 2012.*

15  **Question #6:** The target probability of failure for wind turbine blade given by design standards is generally, very low. Can authors justify why the calculated probability of failure (e.g. the results presented in Fig. 15) is much higher than the target probability of failure given by design standards?

   *AR*  Additional details have been provided which describe a background to the failure of probability analysis and how
20  the analysis is a compounded "worst on worse" case.

**A marked-up version of the paper with the changes follows:**

[revised manuscript text omitted]

---

## Author Response (AR2)

January 29, 2018

Dr. Athanasios Kolios
Associate Editor
Wind Energy Science Journal

5

Dear Dr. Kolios,

Thank you for taking the time to review and comment on our submission. Each of your comments are addressed in the most recently uploaded submission.

1. Both Parts 1 & 2 were published on 19 Dec 2017 and may be found [here] and [here].
10
2. The borders and colors have been adjusted for consistency throughout. Particularly, Figures 1, 4, & 5 for both while 9, 11, 12, 13, 14, & 18 have adjusted only for color.

We appreciate your comments and look forward to the review process. Please do not hesitate to contact us if you have any further questions or concerns.

15   Kind regards,
Jared, Doug, & Trey

Jared W. Nelson, PhD
Assistant Professor of Mechanical Engineering
20   101 Resnick Engineering Hall
1 Hawk Drive
New Paltz NY 12561
Phone: (845) 257-3887
Fax: (845) 257-3730
25   [nelsonj@newpaltz.edu]
[www.newpaltz.edu]